# Hot Spot Mutagenesis Improves the Functional Expression of Unique Mammalian Odorant Receptors

**DOI:** 10.3390/ijms23010277

**Published:** 2021-12-28

**Authors:** Yosuke Fukutani, Yuko Nakamura, Nonoko Muto, Shunta Miyanaga, Reina Kanemaki, Kentaro Ikegami, Keiichi Noguchi, Ikuroh Ohsawa, Hiroaki Matsunami, Masafumi Yohda

**Affiliations:** 1Department of Biotechnology and Life Science, Tokyo University of Agriculture and Technology, Tokyo 184-8588, Japan; yuko.nakamura@yohda.net (Y.N.); nonoko.muto@yohda.net (N.M.); shunta.miyanaga@yohda.net (S.M.); reina.kanemaki@yohda.net (R.K.); kentaro.ikegami@yohda.net (K.I.); yohda@cc.tuat.ac.jp (M.Y.); 2Instrumentation Analysis Center, Tokyo University of Agriculture and Technology, Tokyo 184-8588, Japan; knoguchi@cc.tuat.ac.jp; 3Biological Process of Aging, Tokyo Metropolitan Institute of Gerontology, Tokyo 173-0015, Japan; iohsawa@tmig.or.jp; 4Department of Molecular Genetics and Microbiology, Duke University Medical Center, Durham, NC 27710, USA; matsu004@mc.duke.edu

**Keywords:** odorant receptor, chemosensory, membrane traffic, heterologous expression

## Abstract

Vertebrate animals detect odors through olfactory receptors (ORs), members of the G protein-coupled receptor (GPCR) family. Due to the difficulty in the heterologous expression of ORs, studies of their odor molecule recognition mechanisms have progressed poorly. Functional expression of most ORs in heterologous cells requires the co-expression of their chaperone proteins, receptor transporting proteins (RTPs). Yet, some ORs were found to be functionally expressed without the support of RTP (RTP-independent ORs). In this study, we investigated whether amino acid residues highly conserved among RTP-independent ORs improve the functional expression of ORs in heterologous cells. We found that a single amino acid substitution at one of two sites (N_BW_3.39 and 3.43) in their conserved residues (E and L, respectively) significantly improved the functional expression of ORs in heterologous cells. E^3.39^ and L^3.43^ also enhanced the membrane expression of RTP-dependent ORs in the absence of RTP. These changes did not alter the odorant responsiveness of the tested ORs. Our results showed that specific sites within transmembrane domains regulate the membrane expression of some ORs.

## 1. Introduction

Olfaction is an essential sensation for animals to continue carrying out their daily life activities while experiencing changes in the surrounding environment. Mammalian olfactory receptors (ORs) are expressed on the cell surface of olfactory sensory neurons (OSNs) to detect and discriminate the vast number of odors in the environment [1]. ORs form the largest and most diverse family of G protein-coupled receptors (GPCRs) [2]. There are more than 400 and 1200 functional ORs in the olfactory sensory neurons of humans and mice, respectively [3,4,5]. It is thought that this diverse odor recognition of ORs has made it possible for olfaction to discriminate odors, which are a mixture of various volatile molecules, with high sensitivity [6,7]. A major difficulty in characterizing the function of these mammalian ORs is that most ORs are not exported to the plasma membrane and are instead retained in the endoplasmic reticulum (ER) in nonolfactory cells, making in vitro characterization extremely challenging [2]. Consequently, functional analysis of ORs is far behind that of other canonical Class A GPCRs. Therefore, the flexible and selective odor molecule recognition mechanism of ORs is almost unknown. To fill this gap in knowledge, it is necessary to develop a method for improving OR expression without changing the function of ORs. The use of a Rho tag or IL6 tag attached to the N-terminus of ORs has been suggested as a method due to the technology’s ability to promote the enhancement of surface expression of ORs [8,9,10]. Additionally, co-expression of chaperone proteins, Hsp70t or Ric-8B, is effective for the functional expression of some ORs [11,12]. However, both modifications only enhance the expression of a limited number of ORs.

Heterologous expression of many ORs is achieved with the assistance of receptor transporting protein (RTP) 1S and RTP2 [13,14,15,16,17]. These are single transmembrane proteins, a class of chaperones, that mediate the transport of ORs into the plasma membrane. However, the detailed mechanism of the membrane trafficking of ORs by RTPs is unknown. Interestingly, a small subset of ORs can localize to the cell membrane without RTP1 and RTP2 in heterologous cells and function even in double knockout mice lacking RTP1S and RTP2 [18]. We previously demonstrated the importance of a specific amino acid residue within transmembrane domain 4 (G^4.53^) for the functional expression of OR in heterologous cells. Poor cell surface expression seems to correlate with structural instability [19].

This study investigated whether mammalian OR expression could be improved by replacing an amino acid with one that is highly conserved in a well-expressed OR (RTP-independent OR). We also tried to identify important amino acid residues that contribute to the functional expression of mammalian ORs.

## 2. Results

### 2.1. Screening of the Amino Acid Mutations That Enhance OR Expression

We hypothesized that highly conserved amino acid residues in RTP-independent ORs play essential roles in OR trafficking in heterologous cells. Our previous study showed that amino acid residues at 66 sites are associated with RTP independence [19]. Among them, the conserved amino acid residues at 26 sites appeared more frequently in RTP-independent ORs than in all intact 1093 mouse ORs. Their positions were dispersed throughout the OR sequence (Figure 1A and Appendix A). We hypothesized that these ORs that have non-conserved amino acid residues at these positions might show higher levels of expression when these residues are changed to conserved ones.

We tested this hypothesis by constructing 40 mutant ORs, transiently expressing them in HEK293T cells and analyzing their cell surface expression by FACS. We first tested 14 sites out of the 26 in 14 RTP-independent ORs (Figure 1A Red circle). Among them, Olfr544 (also known as OR-S6), D115E (N_BW_3.39), Olfr194_128H (IC2), and Olfr78_Q297E (C-term) exhibited a significant increase in surface expression (Figure 1B, Appendix A).

We next tested whether the changes to the conserved residues in these sites improve the expression of RTP-dependent ORs. In the same manner, we changed a non-conserved amino acid residue to a conserved residue in 11 RTP-dependent ORs. Ninety-seven single amino acid mutants covering all 26 sites were constructed and expressed in HEK293T cells without RTP1S coexpression. Two mutants (Olfr992D111E (N_BW_3.39) and Olfr982Y120L (N_BW_3.43)) showed enhanced plasma membrane localization by more than 30% (Figure 1C, Appendix A). Since N_BW_3.39 and N_BW_3.43 are in proximity in the same transmembrane helix, interactions between them may contribute to the expression of ORs. Based on these screenings with both RTP-dependent and RTP-independent ORs, we decided to focus on our study on residues within transmembrane domain 3.

### 2.2. Improvement of Surface Expression by the 3.39E and 3.43L Mutations in Many ORs

In 1093 intact mouse ORs registered in the HORDE database [21], E(glutamic acid) accounted for 85.4% (933/1093) of N_BW_ 3.39, and L(leucine) accounted for 93.3% (1020/1093) of N_BW_ 3.43 (Figure 2A,B). To examine whether mutations of nonconserved N_BW_3.39 and N_BW_ 3.43 residues to E (3.39E mutation) and L (3.43L mutation) are effective for expressing various ORs, we applied the mutations to mouse ORs with known ligands. We made 3.39E mutants for six ORs (Olfr339, Olfr992, Olfr1352, Olfr1353, Olfr1377, and Olfr960) and 3.43L mutants for five ORs (Olfr62, Olfr982, Olfr978, Ofr979, and Olfr960) and measured the mutant expression level on the cell surface (Figure 2A–C). Surface expression on heterologous cells was improved for all single mutants except Olfe1352 D3.39E in the co-expression of RTP1S. Furthermore, 6 ORs were localized on cell membranes without co-expression of RTP1S after introducing mutations of 3.39E or 3.43L (Figure 2D). The cell surface expression of Olfr960, which has both L3.39 and Y3.43, was not improved by either mutation (L3.39E and/or Y3.43L). However, the effect of RTP1S co-expression on the mutants was higher than that on the wild type (Figure 2C,D)

### 2.3. Olfr544 D3.39E Showed Significantly High Expression without a Change in Ligand Selectivity

The single amino acid mutation D3.39E significantly improved the membrane expression of Olfr544. The expression level of Olfr544D115E^BW3.39^ alone was significantly higher than with the co-expression of RTP1S (Figure 3A,B). Olfr544 responds to dicarboxylic acids, including nonanedioic acid (azelaic acid) [22]. We tested Olfr544 WT and D3.39E mutant responses to dicarboxylic acids with various carbon chain lengths (Figure 3C). The responses of the D3.39E mutant were higher than those of the WT, correlating with the higher expression level in the mutant. The selectivity for dicarboxylic acids with different carbon chains was not affected by the mutation. The D3.39E mutant responded to concentrations of nonanedioic acid 100 times lower than the WT, with statistical significance (Figure 3D). The ligand selectivity did not change for the tested ligands.

Olfr544 belongs to the Class I OR family, and D3.39 is widely conserved in Olfr544 homologs (Appendix A), suggesting that Olfr544D3.39 appeared relatively early and was subsequently maintained during evolution. When D3.39 of Olfr544 was mutated to various amino acids, the D3.39E mutant showed the highest cell surface expression level among the mutants (Appendix A). The D3.39R and D3.39N mutants showed increased expression but lost responsiveness to their agonist. Since N_BW_3.39 of ORs is located in the sodium ion binding site in Class A GPCRs [23,24], it is plausible that the interaction of glutamate with sodium ions is important for the stability and function of ORs.

To gain insights into the mechanism of the improvement of stability by the N_BW_3.39E mutation, we constructed structural models of Olfr544 variants, wild type, D3.39E (D115E) mutants, and D3.39R (D115R) mutants using AlphaFold 2 [25]. The overall structures of the models were almost identical. The distance between the side chain of N7.49 and the side chain of D3.39 (D115) is 4.5 Å in the wild type. In the D3.39E mutant, the distance between E3.39 and N7.49 was shortened to 3.4 Å to form a hydrogen bond (Figure 3E).

In the model of the D3.39R mutant, the arginine residue of N_BW_3.39 was located in the center of the lumen of the GPCR, similar to other reported Class A GPCRs (Appendix A) [26]. The effects of the D3.39E and D3.39R mutations are related to the coordination and allosteric action of the sodium ion. In other words, this suggests that N_BW_3.39 of ORs also functions in the allosteric movement of ORs as a sodium ion binding site, which is similar to other Class A GPCRs.

### 2.4. Agonist Response of N_BW_3.39E and N_BW_3.43L Mutant ORs

Then, we examined the effects of the N_BW_3.39E or N_BW_3.43L mutation on the ligand responses of various ORs. Most OR mutants showed an improved odor response compared to WT (Figure 4A–C and Appendix A), correlating with the increase in surface expression. However, the Olfr992D3.39E and Olfr62F3.43L mutants did not respond to the original ligands of octanoic acid and 2-coumaranon, respectively (Figure 4D). Since there is a correlation between amounts of OR expression and ligand responsiveness in heterologous cells [17], these results suggest that the mutation sites Olfr992D3.39 and Olfr62F3.43 are likely to play a role in ligand binding.

### 2.5. The Change in Conserved E3.39 and L3.43 Caused a Loss of Function

To investigate whether mutation of the conserved N_BW_3.39E and N_BW_3.43L alters the functional expression of ORs, we constructed E3.39D and L3.43Y mutants. We selected Olfr1508 (agonist: Cu^2+^ ion) and Olfr733 (agonist: hexanal), which are highly expressed in the cell membrane without the assistance of RTPs [19,27]. The E3.39D and L3.43Y mutations significantly reduced their cell surface expression and ligand response ability (Figure 5).

Although co-expression of RTP1S could rescue the functional expression of L3.43Y mutants, the E3.39D mutant was not recovered. These results suggest that both N_BW_3.39 and N_BW_3.43 of ORs are important residues for the appropriate expression of Ors.

### 2.6. N_BW_ 3.39E and N_BW_ 3.43L Mutations Are also Effective in Human Ors

We examined the effects of the 3.39E and 3.43L mutations on human Ors. In human Ors, E(Glutaminc acid) accounted for 81% (315/390) of N_Bw_3.39 and L(Leucine) accounted for 92.5% (361/390) of N_Bw_3.43. (Figure 6A,B). Ors without conserved E3.39 tend to be clustered in a few families, suggesting evolutionary advantages in these families. On the other hand, the Ors whose 3.43 was not L were dispersed throughout the phylogenetic tree, and many of them were pseudogenes. Then, we constructed 3.39E and 3.43L mutants and examined their surface expression and ligand response (Figure 6 and Appendix A). OR7D4 is one of the characteristic Ors in which human perception of androstenone changes dramatically depending on genetic variants [28]. The D3.39E mutation significantly increased the cell surface expression of OR7D4, and the detection limit of androstenone (5α-androst-16-en-3-one) was improved to less than 1 µM (Figure 6C). A significant increase was also observed in OR10G7 Y3.43L, which was expressed without co-expression of RTP1S (Figure 6D). This mutant also detected a significant response against Eugenol even at lower stimulus concentrations than the wild type [29]. Mutations in the N_BW_3.39 and 3.43 amino acids can be effective ways to improve the functional expression of human ORs. Taken together, the amino acids located at N_BW_3.39 and 3.43 are involved in regulating the expression of mammalian ORs.

## 3. Discussion

Animals can detect various odorants in the environment through pattern recognition of the response by hundreds of ORs [7,30]. We presented a hypothesis that olfactory neurons acquired a high level of evolutionally capacitance to allow rapid evolution of ORs [19]. This rapid evolution allows ORs to respond to structurally diverse environmental odor molecules but compromises conformational stability, resulting in difficulties in functional expression of ORs in non-olfactory neurons [19]. In this study, from the amino acid residues conserved among the RTP-independent ORs with high conformational stability, we found that N_BW_3.39 and 3.43 in the third transmembrane domain are related to OR expression. The 3.39E and 3.43L substitutions were effective in improving the membrane localization of multiple ORs in mice and humans. Curiously, conservation of N_BW_3.39E and N_BW_3.43L in human ORs is lower than that in mouse ORs. This may, in part, explain why expression levels of the human ORs are generally lower than those of mouse ORs in heterologous cell expression systems. 3.39E mutations of the ORs containing D at N_BW_3.39 often enhanced membrane expression (7/10 in tested ORs), while the 3.39E mutation was less effective for ORs with other amino acids at N_BW_3.39, such as Q (1/4 in tested ORs). Both N_BW_3.39 and 3.43 are in the transmembrane helices and face the same direction. Interestingly, N_BW_4.53, which affects the RTP dependency of Olfr539, also faces towards N_BW_3.39 and N_BW_3.43 [19]. Together, the luminal environment of each transmembrane helix may be important for the stability of ORs and their associated membrane localization. This may occur via the stabilization of membrane packing. Residues in both TM3 and TM4 of ORs may act in combination to stabilize membrane packing. This work successfully identified two specific residues to facilitate the expression of ORs. The expression levels of the human ORs were lower than those of mouse ORs in heterologous cell expression systems. By introducing these mutations, detailed analysis of orphan ORs, which could not be analyzed due to its low expression level, may be achieved.

We previously report that the introduction of arginine to specific residues predicted to improve thermal stability also improved the functional expression of consensus ORs [19]. It has also been reported that the functional expression of OR was improved by introducing mutations to transmembrane domains other than TM3 and TM4 and the C-terminal region [31,32]. These methods that have been effective to some ORs appears to be ineffective with other ORs. As a consequence, no single method can be adapted to improve the expression of all mammalian ORs. There may be different solutions to the OR trafficking problem depending on the individual ORs. However, the solutions can be broadly categorized into a small number of methods, one of which would be to stabilize membrane packing by adding appropriate mutations.

N_BW_3.39 and N_BW_3.43 are located in the vicinity of the sodium ion binding site in other Class A GPCRs. N_BW_3.39 is frequently mutated to increase the production yield for crystallization of GPCRs [26,33]. The 8 amino acids out of the 15 sodium-binding sites reported are different from those of ORs to the canonical Class A GPCR (Appendix A), and the relationship between the functional activity of the ORs and the sodium ion has not been reported thus far. This difference in amino acid conservation of the sodium ion-binding site may mean that the origin of the mammalian ORs is fundamentally different from other GPCRs.

It is very likely that sodium ions are coordinated in the vicinity of N_BW_3.39 and act allosterically, the same as in other GPCRs. While the three-dimensional structures of many Class A GPCRs have been reported, none of the ORs have been solved. Reports of ligand docking analysis and MD simulation of ORs using a model structure are increasing [34,35,36,37]. The progress of structure prediction technologies including AlphaFold 2 has been remarkable [25]. Our study identifying important residues that contribute to the functional expression of ORs will help lead to the elucidation of the functional mechanism of ORs and the development of high-sensitivity odor sensing technology using mammalian ORs.

The introduction of the 3.39E or 3.43 L mutation significantly affected the functional expression of ORs. Most mutant ORs retained odorant selectivity except for Olfr544 D3.39R, which caused a loss of ligand response while improving membrane localization. Although the mutations (N_BW_3.39E and N_BW_3.43L) we found in this study are applicable to a small number of ORs (approximately 19% and 7.5% of human ORs, respectively), the ORs may have evolved to have unique characteristics in odor detection in some cases. Future experiments can address this hypothesis by expanding the number of ORs and odorants tested using N_BW_3.39E and N_BW_3.43L mutations. Currently, however, the majority of mammalian ORs are still poorly expressed even with the aid of RTPs, suggesting that pieces are still missing for the stable expression of ORs in heterologous cells.

## 4. Materials and Methods

### 4.1. DNA and Vector Preparation

Open reading frames of OR genes were subcloned into pCI (Promega, Madison, WI, USA) with a rhodopsin tag at the N-terminus. To generate mutants of ORs, DNA fragments of OR genes were amplified by Phusion polymerase (Thermo Fisher Scientific, Waltham, MA, USA) and PrimeStar MAX polymerase (Takara bio, Shiga, Japan). The fragments were mixed and amplified by PCR to obtain full sequences. All plasmid DNA was purified by NucleoSpin plasmid transfection grade (MACHEREY-NAGEL GmbH & Co, Düren, Deutschland). Other expression vectors were the same as used in a previous study [17]. All plasmid sequences were verified using Sanger sequencing.

### 4.2. Cell Culture

HEK293T and Hana 3A [13] cells were grown in Minimal Essential Medium (MEM) containing 10% FBS (vol/vol) with penicillin-streptomycin and amphotericin B. Hana 3A cells were authenticated using polymorphic short tandem repeats (STRs) at the Duke DNA Analysis Facility using GenePrint 10 (Promega) and shown to share profiles with the reference (ATCC). All cell lines were incubated at 37 °C, saturating humidity and 5% CO_2_. No mycoplasma infection was detected in any cell culture.

### 4.3. Flow Cytometry Analyses

HEK293T cells were grown to confluence, resuspended, and seeded onto 35 mm plates at 25% confluency. The cells were cultured overnight. The Rho-tagged OR in the plasmid pCI and GFP expression vector were transfected using Viafect transfection reagent (Promega). After 18–24 h, the cells were resuspended in a cell stripper (Corning, NY, USA) and then kept on ice. The cells were spun down at 4 °C and resuspended in PBS containing 15 mM NaN_3_ and 2% FBS to wash away the cell stripper. They were then incubated with a primary antibody mouse anti-Rho4D2 mouse antibody (Merck-Millipore, Burlington, MA, USA), washed, and then stained with phycoerythrin (PE)-conjugated donkey anti-mouse F(ab’)₂ fragment antibody (Abcam, Cambridge, UK) in the dark. To stain dead cells, 7-amino-actinomycin D (Merck-Millipore, Burlington, MA, USA) was added. The cells were analyzed using BD FACSCanto II FACS and BD LSRFortessa with gating allowing for GFP-positive, single, spherical, viable cells, and the measured PE fluorescence intensities were analyzed and visualized using FlowJo.

### 4.4. Immunocytochemistry

Live-cell surface staining was performed as described previously [38]. The primary antibody used was mouse anti-rhodopsin 4D2 (Merck-Millipore). The secondary antibodies used were Cy3-conjugated anti-mouse IgG antibody (Thermo Fisher Scientific, Waltham, MA, USA). After antibody staining, the cells were fixed in 4% paraformaldehyde, and then the slides were mounted with Mowiol and visualized by a fluorescence microscopy M1 imager (Carl Zeiss, Oberkochen, Deutschland).

### 4.5. Dual Luciferase Reporter Gene Assay

All odorants were purchased from FUJIFILM Wako Chemicals (Osaka, Japan) and TCI chemicals (Tokyo, Japan). The Dual-Glo Luciferase Assay (Promega) was used to determine the activities of firefly and Renilla luciferase in Hana3A cells as previously described [38]. Briefly, firefly luciferase, driven by a cAMP response element promoter (CRE-Luc; Stratagene), was used to measure the OR activation levels. For each well of a 96-well plate, 5 ng SV40-RL, 10 ng CRE-Luc, 5 ng mouse RTP1, 2.5 ng M3 receptor 3, and 5 ng of Rho-tagged receptor plasmid DNA were transfected. Normalized activity for each well was further calculated as (Luc)/(Rluc). Luc and Rluc are the luminescence of firefly luciferase and Renilla luminescence, respectively. The basal activity was averaged from six wells in the absence of odorants and further corrected by subtracting that of the control empty vector. Odorant-induced activity was averaged from at least three wells and further corrected by subtracting the basal activity of that receptor. Odorant-induced responses were normalized to that of WT. EC50 value was calculated using GraphPad Prism software and data shown in Appendix A.

### 4.6. Statistical Analysis

Multiple comparisons were performed using one-way or two-way analysis of variance (ANOVA) using GraphPad Prism. Student’s t tests were performed using the built-in function in Microsoft Excel. The average is shown as the mean ± standard errors.

## Figures and Tables

**Figure 1 ijms-23-00277-f001:**
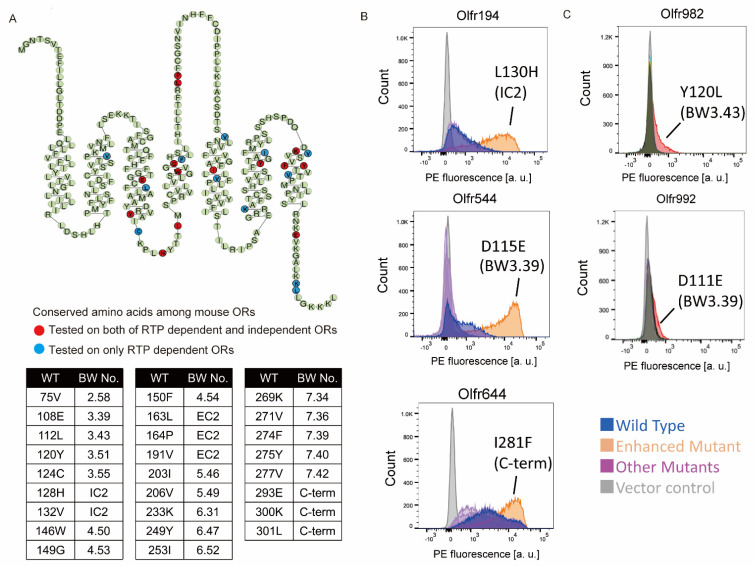
Identification of mutation residues enhancing OR expression. (**A**) Snake plot showing the consensus sequence of mouse ORs. Circle: Amino acid site that constructed the mutant. Red: tested on both of RTP dependent and independent ORs, Blue: tested on RTP dependent ORs only. The table shows the mutation sites and BW (Ballesteros-Weinstein) number [20] tested in this study. (**B**,**C**) The cell-surface expression of the RTP independent OR (Olfr194, Olfr544 and Olfr644) and RTP dependent ORs (Olfr982 and Olfr992), respectively. Each mutant was transfected into HEK293T cells, and the cell surface expression level was measured. X-axis: PE fluorescence, Y-axis: cell number.

**Figure 2 ijms-23-00277-f002:**
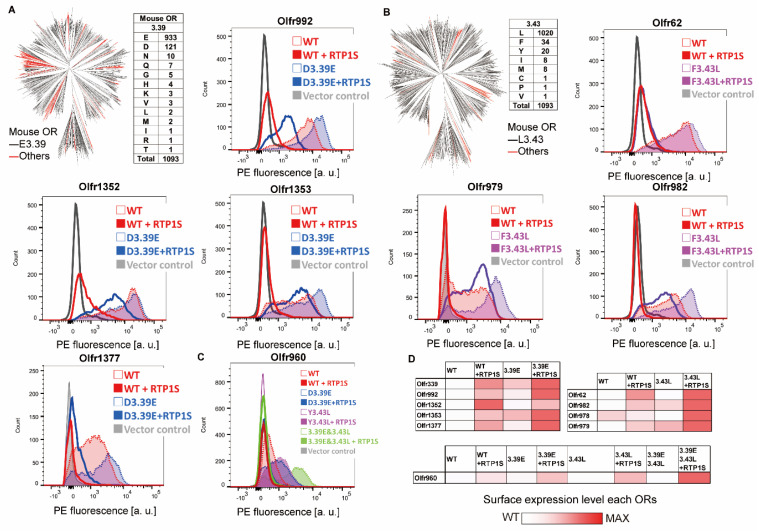
Effect of the 3.39E and 3.43L mutants on mouse ORs. (**A**) Phylogenic tree of mouse ORs showing the E3.39 or others. Red indicates an OR in which the residue of N_BW_3.39 is not E (Glu) in 1093 intact mouse ORs. Cell-surface expression of the Rho-tagged ORs and 3.39E mutants. Each mutant was transfected into HEK293T cells, and the cell surface expression level was measured. X-axis: PE fluorescence, Y-axis: cell number. (**B**) Phylogenic tree of mouse ORs showing the L3.43 or others. Red indicates an OR in which the residue of N_BW_3.43 is not L (Leu). Cell-surface expression of the Rho-tagged ORs and 3.43L mutants. (**C**) Cell-surface expression of Rho-tagged Ofr960 and its mutants. (**D**) The values of the effect of mutations of 3.39E and 3.43L were used as the max intensity (red) and wild type (white) in each OR, respectively.

**Figure 3 ijms-23-00277-f003:**
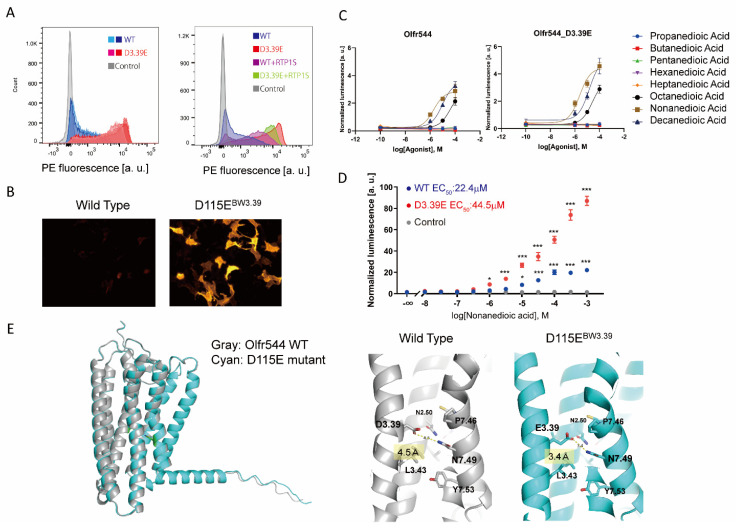
Detailed analysis of Olfr544 D115E^BW3.39^ (**A**) Cell-surface expression of the Rho-tagged ORs and D3.39E mutants. Each mutant was transfected into HEK293T cells, and the cell surface expression level was measured. X-axis: PE fluorescence, Y-axis: cell number. Left: Comparison between the WT and the D3.39E mutant. The results from two independent experiments are shown in the graph. Right: The effect of RTP1S co-expression of Olfr544 and D3.39E mutant. (**B**) Immunocytochemical image of Olfr544 and the D3.39E mutant stained with anti-Rho4D2 mouse antibody and PE-conjugated anti-mouse IgG goat antibody. (**C**) Agonist selectivity of Olfr544 and the D3.39E mutant for various dicarboxylic acids. Error bars indicate s.e.m (n = 3). (**D**) Dose response curve of Olfr544 and the D3.39E mutant to nonanedioic acid. Error bars indicate s.e.m (n = 3). Multiple comparisons were performed using one-way ANOVA followed by Dunnett’s test (* *p* < 0.05, *** *p* < 0.001). (**E**) Structural model of Olfr544 and the D3.39E mutant using Alphafold 2 prediction. It was predicted that the distance between the residue of 3.39 and N7.49 was 4.5 Å in the WT (D3.39) and 3.4 Å in the E3.39 mutant.

**Figure 4 ijms-23-00277-f004:**
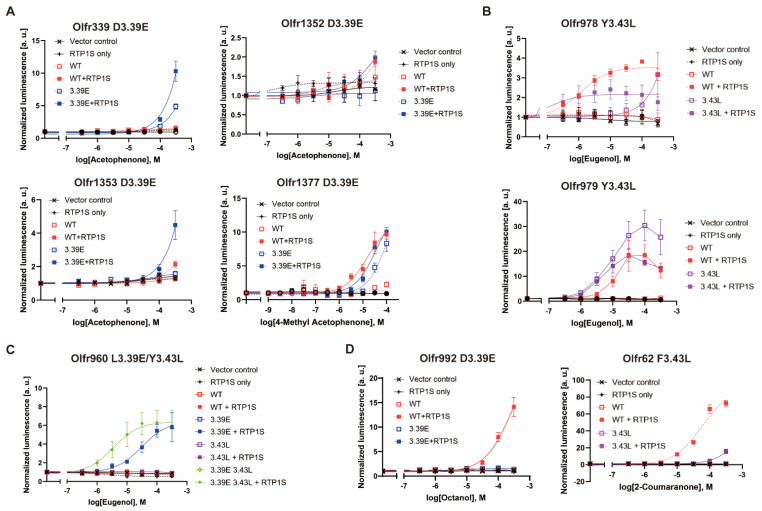
Dose-response curve of each mouse OR mutant. (**A**) D3.39E mutants of Olfr339, Olfr1352, Olfr1353 and Olfr1377, (**B**) Y3.43L mutants of Olfr978 and Olfr979, (**C**) Olfr960 L3.39E, Y3.43L and double mutants, (**D**) ORs with reduced agonist response due to mutations. Olfr992D3.39E against octanol and Olfr62F3.43L against 2-coumaranone. All constructs, including the vector control, were also tested under RTP1S co-expression conditions. Error bars indicate s.e.m (n = 3).

**Figure 5 ijms-23-00277-f005:**
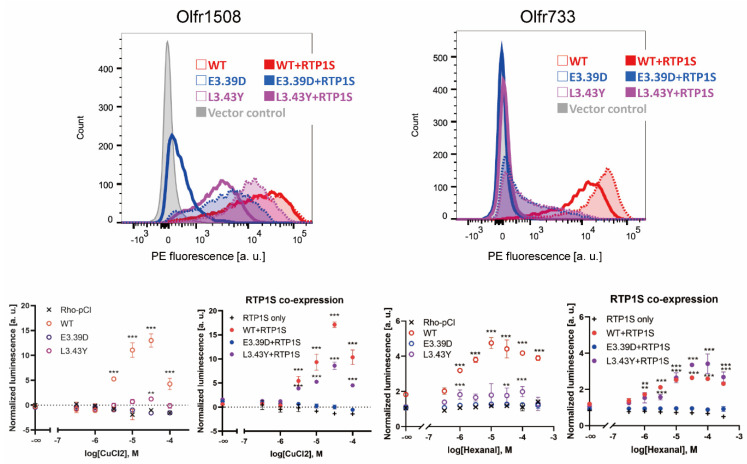
The change in conserved E3.39 and L3.43 led to a loss of function. (**Top**) The cell-surface expression of the Rho-tagged wild-type, E3.39D and L3.43Y mutants of Olfr1508 and Olfr733. Each OR was transfected into HEK293T cells, and the cell surface expression level was measured. X-axis: PE fluorescence, Y-axis: cell number. Left graph: Comparison between the WT and D3.39E mutant. The results from two independent experiments are shown in the graph. Right: The effect of RTP1S co-expression on them. (**Bottom**) Dose-response curve of Olfr1508 and Olfr733 mutants to CuCl2 (Cu2+) and hexanal, respectively. Error bars indicate s.e.m (n = 3). Multiple comparisons were performed using one-way ANOVA followed by Dunnett’s test (** *p* < 0.01, *** *p* < 0.001).

**Figure 6 ijms-23-00277-f006:**
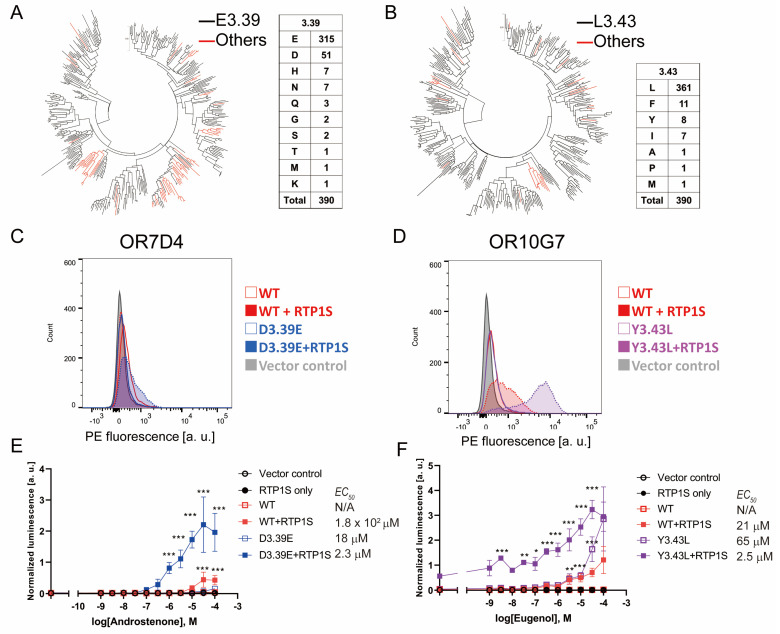
Effects of 3.39E and 3.43L mutants on human ORs (**A**,**B**) Phylogenic tree of mouse ORs showing the E3.39 or others, or L3.43 or others in 390 intact human ORs registered in the HORDE database. (**C**,**D**) Cell-surface expression of the Rho-tagged ORs and 3.39E mutants. Each mutant was transfected into HEK293T cells, and the cell surface expression level was measured. X-axis: PE fluorescence, Y-axis: cell number. (**E**,**F**) Dose response curves of OR7D4 (WT and muant) and OR10G7 (WT and mutant) to androstenone and eugenol, respectively. Error bars indicate s.e.m (n = 3). Multiple comparisons were performed using one-way ANOVA followed by Dunnett’s test (* *p* < 0.05, ** *p* < 0.01, *** *p* < 0.001).

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
