# Peer review of "Hot Spot Mutagenesis Improves the Functional Expression of Unique Mammalian Odorant Receptors"

_ijms, 2021, doi:10.3390/ijms23010277_

Round 1

Reviewer 1 Report

Authors investigated whether amino acid residues highly conserved among RTP-independent ORs improve the functional expression of ORs. They claim that only a single amino acid mutation at one of two sites (NBW3.39 and 3.43) in their conserved sites (E and L, respectively) significantly improved the functional expression of ORs in heterologous cells. E3.39 and L3.43 also enhanced the membrane expression of RTP-dependent ORs in the absence of RTP. 

The claims made in this manuscript are well supported by the data. Overall this is a significant advance in the field. I recommend the publication of this manuscript in the current form.

Author Response

Thank you very much for your review and kindly comments.

Reviewer 2 Report

In this work, Fukutani and colleagues analyzed the role played by specific amino acids in the expression of ORs in the cell membrane of HEK293T cells. Amino acid positions that are conserved in ORs that do not depend on the chaperone proteins RTPs 1 and 2 to reach the plasma membrane were analyzed in this study. A large number of mutant ORs were constructed, expressed in HEK293T cells and analyzed in terms of surface expression and responses to ligands.  

The authors found that two amino acid positions in TM3 play a role in surface expression of the ORs. Many of the mutant ORs carrying these amino acids show increased membrane expression, without showing altered ligand specificities.

The approach and results are interesting and important to the field. Unfortunately, the article is very poorly written, the results are presented in an unorganized fashion, making it hard to follow. There are several problems with the presentation of the figures and figure legends. There are also several problems with the language as well. Examples of these are given bellow:

1-Resolution of figure 1A is too low, cannot read the amino acids. Define BW in the legend of figure 1 and explain what it means in text, since it is relevant to the understanding of the OR mutants.

45 conserved aas are shown in red figure 1a, the table shows 25 aa, and in the text is said that ‘Our previous study showed that 66 amino acid residues contribute to RTP independence [17]’. Which ones were the sites that were tested? The 25 in the table? The 45 in the snake plot? Please clarify. Would help if the amino acid positions tested in the study would be colored differently, to make it easier to localize them in the OR snake plot. For example, the 14 positions tested in the RTP independent ORs, and the positions tested in the RTP dependent ORs.

For example, page 3 line 85: ‘Ninety-seven single amino acid mutants covering all of the candidate sites were constructed and expressed in HEK293T cells without RTP1S coexpression’. Means: ‘Ninety-seven single amino acid mutants covering all of the 45? candidate sites were constructed and expressed in HEK293T cells without RTP1S coexpression’

2- The sentence: ‘conserved amino acids in RTP independent ORs than in RTP dependent ORs’ sounds weird, was it meant: ‘conserved amino acids in RTP independent ORs when compared to RTP dependent ORs’? or just ‘conserved amino acids in RTP independent ORs’.

3- Page 2 line 69: ‘Their positions were dispersed throughout the ORs’ replace by ‘Their positions were dispersed throughout the OR sequences’

4-Page 3 line 89:’ Two mutants (Olfr992D111E (NBW3.39) and Olfr982Y120L (NBW3.43)) showed enhanced plasma membrane localization by more than 30% (Fig. 1C, S1 and S3).’

There is no Fig. 1 C in figure 1. (at least not labeled as such). And no corresponding legend to the experiment with mutants Olfr992 and 982, so it is hard to follow these results. Are these positions in the Table in fig 1A? Y120 is, but what about D111? Why is the BW different in 120Y and Y120L?  explain ‘new’.

5- Colors in figure S3 seem to have faded away.

6- Page 3, line 90: ‘helix with the same direction of the side chains’, what exactly is meant by this sentence?

7- Page 4, line 102: HORDE.

8- Page 4, line 104: ‘accounted for 93.3% (1020/1093) ‘complete the sentence: 93.3% of what?. Same for page 8 line 192.

9- Figure 3A: labels in the graph on the left in Fig 3A should be corrected (delete the dark blue and dark red?). I understood from the legend that in this case there is no addition of RTP1, but the light red color appears in the graph? Or the dark and light colors represent different experiments without RTP1? Should use different colors then, otherwise the reader will be confused. Fig 3B: why is labeling in the WT so much lower than the mutant (if this OR is RTP independent?)

10- Page 4 line 126: ‘Therefore, the D3.39E mutation improves its sensitivity for its ligand by increasing its expression level without altering its ligand selectivity.’ The ligand selectivity did not change, at least for these tested ligands.

11- Figure 4 is low resolution. Legend of figure 4 should be rewritten.

12- EC50s should be provided for all functional assays.

Reviewer 3 Report

The presented work contains a lot of interesting information. Amino acid mutations can be effective ways to improve the functional expression of human ORs. The research was planned correctly and the obtained results do not raise any objections. I have a few comments below and I ask you to answer them.

line 34-40,  GPCRs roles in vertebrate and invertebrate olfaction should be more clearly presented in the introduction section, alongside with recent literature where they can be utilized:  10.3389/fncel.2020.00067,  10.1016/j.trac.2021.116330, 10.1016/j.bios.2019.111923 

line 60-62, it is not clear what the scientific novelty of this work is, please underline this novelty.

line 77-82, How was transfection of cell cultures performed?

Line 102, I believe it's a HORDE database.

line 166-167, and what else might play a role in ligand binding to the OR? Do the authors have any hypothesis?

line 294-300, Which luciferase assay system was used in this study?

Round 2

Reviewer 3 Report

I have no more remarks. The work has been improved, I recommend it for further stages of the evaluation.